# Detecting Incorrect Visual Demonstrations for Improved Policy Learning

**Mostafa Hussein**
Department of Computer Sciences
University of New Hampshire
mostafa.hussein@unh.edu

**Momotaz Begum**
Department of Computer Sciences
University of New Hampshire
momotaz.begum@unh.edu

**Abstract:** Learning tasks only from raw video demonstrations is the current state of the art in robotics visual imitation learning research. The implicit assumption here is that all video demonstrations show an optimal/sub-optimal way of performing the task. What if that is not true? What if one or more videos show a wrong way of executing the task? A task policy learned from such incorrect demonstrations can be potentially unsafe for robots and humans. It is therefore important to analyze the video demonstrations for correctness before handing them over to the policy learning algorithm. This is a challenging task, especially due to the very large state space. This paper proposes a framework to autonomously detect incorrect video demonstrations of sequential tasks consisting of several sub-tasks. We analyze the demonstration pool to identify video(s) for which task-features follow a 'disruptive' sequence. We analyze entropy to measure this disruption and – through solving a *minmax* problem – assign poor weights to incorrect videos. We evaluated the framework with two real-world video datasets: our custom-designed *Tea-Making with a YuMi robot* and the publicly available *50-Salads*. Experimental results show the effectiveness of the proposed framework in detecting incorrect video demonstrations even when they make up $40\%$ of the demonstration set. We also show that various state-of-the-art imitation learning algorithms learn a better policy when incorrect demonstrations are discarded from the training pool.

**Keywords:** Imitation Learning, Visual Demonstrations, Incorrect Demonstrations

## 1 Introduction

Visual imitation learning (VIL) [1, 2, 3, 4] – where tasks are learned from raw video demonstrations– is a promising way for lay users to teach robots new skills in natural settings. VIL however is challenging, especially due to very high dimensional state space. VIL algorithms are generally evaluated in controlled laboratory settings [4, 5, 6, 7, 8, 9] or simulated environments [10, 11, 12, 13, 14], where the assumption that all video demonstrations are optimal/ sub-optimal is possible to maintain. In reality, lay users will make inadvertent errors while demonstrating a task due to reasons including fatigue, presence of distractions, and lack of knowledge on robot learning mechanism. For example, it is not unusual to forget to turn-off the oven once or twice – e.g., because the phone rang – out of the ten times a user was showing the robot how to make a cup of tea. Unless a robotics expert carefully curates the videos, a policy learning algorithm can learn a potentially unsafe tea-making policy – that it is ok to keep the oven on – from these video demonstrations. It is therefore important to automatically analyze video demonstrations for correctness before policy learning takes place. This paper proposes a framework for detecting **incorrect video demonstrations** [1]. Many real-world tasks that a service robot is expected to learn from humans sequential tasks consisting of several sub-tasks, e.g. following a recipe for cooking, tea-making, preparing a dinner table, etc. The proposed framework therefore deals only with videos of sequential tasks.

The state-of-the-art approaches for detecting incorrect demonstrations [15, 16, 17] – almost all of which involve non-visual demonstrations – learn weights to assign importance to different demonstrations. Existing algorithms however come with restrictive assumptions – such as, expert-labeling of incorrect demonstrations [15, 18, 19], using a simulator to generate more data for training

---

[1]We define incorrect demonstrations as the ones that do not follow the standard task definition and are structurally wrong. For example, in the context of a *tea-making* task, an incorrect demonstration would be the one where the demonstrator did not add water or tea-bag to the cup.

6th Conference on Robot Learning (CoRL 2022), Auckland, New Zealand.

[16, 20, 17, 21] – which nullify the core appeal of VIL that a lay user's only responsibility is to show how to do the task, preferably a handful of times; the onus is on the algorithms to learn an accurate policy without demanding more data. We bridge this gap by proposing a framework for autonomous detection of incorrect video demonstration [1] without the need for prior labeling of the videos or a simulation for data augmentation.

The proposed framework leverages two simple facts: 1) spatio-temporal features are better representatives of dynamic human activities than spatial-only visual features and 2) individual tasks of a multi-step task – hereafter referred to as sub-tasks – typically maintain some internal structures among themselves, violation of which is an indication of incorrect demonstration [1]. Accordingly, the intuition behind the proposed framework is to identify groups of spatio-temporal features as the representatives of different sub-tasks (Sections 3.1 and 3.2) and analyze the internal consistency of these features overall videos in the demonstration pool (Section 3.3). To achieve that we propose to solve a minmax optimization problem where we choose the model with the maximum entropy (the most uniform) and at the same time minimizing the demonstrations weight so we can achieve a saddle point with the minimum number of demonstrations. Evaluation of the proposed framework on two real-world datasets – publicly available *50-salad* [22] and our custom-designed *tea-making with YuMi robot*[23] – shows promising results on detecting incorrect demonstrations even when they constitute 40% of the entire demonstration pool. We also show that various state-of-the-art IL algorithms learn better policies when incorrect demonstrations are automatically removed using the proposed framework than IL algorithms that learns from sub-optimal/incorrect demonstrations.

## 2 Related Work

The current imitation learning literature does not explicitly discuss the issue of incorrect demonstrations and how they may affect the learned policy. However, there exists a handful of work that offers mechanisms to differentiate optimal, sub-optimal, and noisy demonstrations during the learning process. This section discusses these methods.

The 2IWIL framework [15] requires an expert to label a fraction of the demonstrations as optimal/non-optimal using confidence score. A semi-supervised classifier is then trained to predict the confidence score of any unlabeled demonstration. Finally, these scores are used by a weighted GAIL framework while learning the task policy. To avoid error accumulation in such a two-step learning process, IC-GAIL [15] trains a single framework for generating the confidence score and learning the policy in an end-to-end fashion. The WGAIL [16] introduces a method of assigning weights to imperfect demonstrations in GAIL [24] without imposing extensive prior information. The success of WGAIL [16] however is limited to simulated environments.

The works that are conceptually closest to our proposed work are R-MaxEnt [25, 26], BCND [27], and DEMO-DICE [28]. R-MaxEnt [25, 26] jointly learns weights for different demonstrations and a task policy. The method however requires carefully-crafted task features, which are only achievable through hand-picking, and therefore has been evaluated for tasks with small state space. BCND [27] learns ensemble policies with a weighted BC objective function where the expert demonstrations are sampled from a noisy expert policy. The weights are generated from the policy learned in a previous internal iteration. DEMO-DICE [28] learns from different level of optimality in the demonstrations but through a much robust convex optimization problem. None of these three approaches however discuss how they can be scaled up for large state space, i.e. visual imitation learning.

Several IL algorithms are capable of detecting demonstrations corrupted by a limited amount of random noise [18, 20, 28]. Demonstrations with artificially injected random noise are far more trivial to process than incorrect demonstrations provided by humans in real world [1]. Additionally, some of these works [18, 20] require simulators to generate additional data.

A number of inverse reinforcement learning-based IL algorithms [19, 29] use "failed" demonstration, along with optimal ones, for policy learning. However, they assume that these failed demonstrations are labeled *a priori* by an expert. Although some recent works on offline reinforcement learning achieve high accuracy in policy learning, they can't handle incorrect demonstrations [30, 31, 32].

All algorithms discussed above are designed to work with non-visual demonstrations and can not be scaled-up to deal with extremely large state space (of video demonstrations). None of these methods can detect incorrect demonstrations without prior labeling of training data or access to more data (through a simulator). Furthermore, all existing work on sub-optimal demonstrations deal with continuous control tasks – e.g. Manipulation, Ant, HalfCheetah, etc. – Where a single incorrect action (e.g. a wrong joint-angle ) may not cause the algorithm to eventually learn a completely

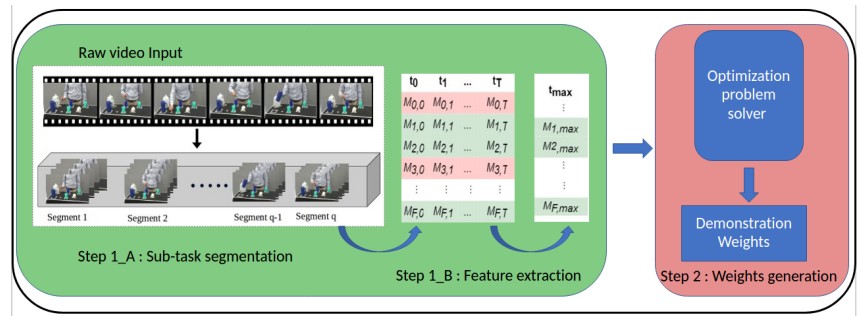

Figure 1: The proposed framework has two steps: the first step (green shaded box, Sections 3.1 and 3.2) takes as an input $n$ raw videos and does sub-task segmentation, generating $q$ segments each with $T$ frames. A features generator extracts $f$ spatio-temporal visual features from each segment, creating a $q \times f$ matrix for each video. The second step (red shaded box, Section 3.3) concatenates all these features and uses them to solve a *minmax* problem which generates $n$ weights, one for each input video.

incorrect policy. In case of multi-step sequential tasks, however, a single incorrect action can lead to an incorrect policy – e.g. not turning off the oven or not adding a tea-bag while making a cup of tea. Our proposed work fills up this void in visual imitation learning research.

## 3 Proposed Framework

The proposed framework identifies incorrect demonstrations of a multi-step sequential task through analyzing the sub-task-representative spatio-temporal features. We leverage the concept of feature expectation matching [33, 34, 35] along with the principle of maximum entropy [36, 37] to learn a stochastic model of the sub-task sequence and generate weights for different demonstrations. Entropy analysis enables us to identify the most consistent group of demonstrations – in terms of sub-task features sequences – and assign higher weights to this group. Note that, multi-step sequential tasks can often be executed in multiple ways – such as making tea with or without milk/sugar – all of which are considered as correct. The proposed model is capable of distinguishing such cases from the case of incorrect demonstrations [1] through generating a distinct weight distribution.

Fig. 1 shows an overview of the proposed framework. The input is a set of $n$ raw video demonstrations and the output is a $n$-dimensional vector representing the weight of each demonstration. The framework calculates these weights in a two-step process. At the first step (green shaded box in Fig. 1) each video is segmented into sub-tasks using any off-the-shelf video segmentation algorithm, e.g. [38] (Section 3.1). Our custom-designed algorithm then extracts spatio-temporal visual features from each sub-task video segments (Section 3.2). These features are representatives of various sub-tasks and are passed to the second step of the framework. During the second step (red shaded box in Fig. 1) these representative features are used to learn a stochastic model of the sub-task sequence which in-turn generates weights for different video demonstrations (Section 3.3).

### 3.1 Sub-task Segmentation

A video segmentation algorithm is used to segment each video in the demonstration set $\mathcal{D}$ into $q$ sub-tasks and generate the start- and end-frame for each sub-task. Here, $q$ is a hyper-parameter for our approach. We employed the semi-supervised MS-TCN++ [38] for this purpose. MS-TCN++ leverages temporal convolution for temporal action segmentation and requires some labeled data for training. MS-TCN++ uses a multi-stage architecture for the temporal action segmentation task. Each stage features a set of dilated temporal convolutions to generate an initial prediction that is refined by the next one. MS-TCN++ requires some training data with the ground truth

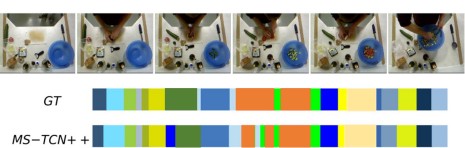

Figure 2: Top row: Sample frames from *50-salad* dataset representing the start of different sub-tasks. Middle row: The ground truth (GT) time-line for each sub-task in this video (colors representing different sub-tasks). Bottom row: Segmentation generated by MS-TCN++.

frame labels. Fig. 2 shows the segmentation of a video from the *50-salad* dataset into $q = 17$ sub-tasks (each sub-task coded with different colors). Note that the only purpose of this segmentation is assigning a consistent label to the set of task-relevant features that will be extracted from the sub-task videos (Section 3.2). The proposed framework is tolerant to typical errors in the existing video segmentation algorithms up to $83\%$ accuracy.

## 3.2 Learning Task-relevant Features

The hallmark of most multi-step sequential tasks is their high-level temporal structure. For example, in the context of a *tea-making* task, *stirring the cup* happens **after** *milk or sugar is added* etc. We employ our own feature generator [39] that can identify and rank a subset of visual features from a video based on their temporal significance. We briefly discuss this feature identification process in this section.

The feature generation happens in three steps: 1) a set of spatial features are extracted from the training videos through any pre-trained convolutional neural networks (CNN) such as VGG-16 [40]. These features, by construction, are not sensitive to temporal content of the video. 2) Extracted spatial features alongside with their automatically extracted time stamp (i.e. when a feature started and stopped appearing in the video) are used to create a graph of temporal relationships among spatial features. This graphical structure – that we name interval temporal relationship (ITR) graph – is used to train a graph convolution network (GCN)[41] which learns the discriminatory temporal features present in the ITR graph. c) These temporally sensitive features are ranked, based on their contribution in classifying the activity in the video, using erasure ranking [42]. Only a set of high-ranked features are used for further analysis. More details are in appendix B.

In summary, each video in the demonstration set $\mathcal{D}$ is automatically segmented into $q$ sub-task segments. For each video segment, a set of ranked features $\{s_j\}_{j=1}^q$ is extracted, $s = \{f_n\}_{n=1}^N$, $N$: the number of features from each segment. These sets, along with the sub-task labels $\{t_j\}_{j=1}^q$, are passed to the next step.

## 3.3 Incorrect Demonstration Detection

We aim to learn a stochastic model $p(t|s)$ that can infer the next sub-task $t \in \mathcal{T}$ given the current state $s \in \mathcal{S}$ and leverage this model to determine a weight $w \in [0, 1]^D$ for each video in the demonstration set $\mathcal{D}$.

To derive the model $p(t|s)$ and the weights $w$ we leverage the principle of maximum entropy (ME) [36, 37] which is proven to choose the most uniform model among a set of constrained models. We use the concept of feature expectation matching (FEM) [33, 34, 35] – a widely popular approach in IL [33, 34, 35] – to define a set of constraints to be observed by $p(t|s)$. In the context of incorrect demonstration detection, the goal is to match the expected value of each feature $\mathbb{E}_{\tilde{p}}[f_i]$ that we identify from the demonstration videos through the feature generator (discussed in Section 3.2) with the calculated expected value $\mathbb{E}_p[f_i]$ generated by the learned model $p(t|s)$. This model $p(t|s)$ will represent the sub-task sequences in the demonstration videos in the most unbiased manner [37].

Along with the model $p(t|s)$, we want to generate a weight $w$ that indicates how well a specific video conform with the learned model. We achieve this by adding to the model-objective the requirement of calculating the minimum weights $w$ that give us the model with the maximum entropy. Using this objective we are forcing the system to generate weights that indicate how consistent a video is with the majority of the videos in the training pool. Poor consistency (lower values of $w$) is an indication of incorrect demonstration. We start with the expected feature definition to formally derive $p(t|s)$:

$$\mathbb{E}_p[f_i] = \sum_{s \in \mathcal{S}} \sum_{t \in \mathcal{T}} \tilde{p}(s)p(t|s)f_i(s,t) \qquad \mathbb{E}_{\tilde{p}}[f_i] = \sum_{s \in \mathcal{S}} \sum_{t \in \mathcal{T}} \tilde{p}(s)\tilde{p}(t|s)f_i(s,t) \qquad (1)$$

Here $\tilde{p}(s)$ is the empirical distribution of the states in $\mathcal{D}$, $\tilde{p}(t|s)$ is the empirical conditional distribution of predicting sub-task $t$ given the state $s$, $f_i(s,t)$ is a feature function whose values are state features corresponding to predicted sub-tasks (Section 3.2). The optimization constraint, according to FEM, is $\mathbb{E}_p[f_i] - \mathbb{E}_{\tilde{p}}[f_i] = 0$. This formalism considers the training set $\mathcal{D}$ as one monolithic whole. Our goal however is to consider each video-demonstration separately and generate a weight $w \in [0, 1]$ for each of them. We therefore redefine the distributions terms and parameterized them by demonstration $d$ and weight $w$ as follows:

$$\tilde{p}_w(t|s) = \frac{1}{M} \sum_{d=1}^D w_d \cdot \tilde{p}(t|s,d) \qquad , \tilde{p}_w(s) = \frac{1}{M} \sum_{d=1}^D w_d \cdot \tilde{p}(s,d)$$

Here $D$ is the total number of demonstrations and $M$, which should be $\sum_{d=1}^D w_d = M$, is the minimum number of demonstrations that we can trust in a given set. Through the experimental results, we can see that the algorithm is not sensitive to the value of $M$. Usually, we can set this number to be $M = D/2$ since a dataset where more than half of the videos have incorrect demonstrations is not suitable for learning.

$$\min_{w \in \mathbb{R}^D} \max_{p(t|s) \in \mathbb{R}^{S \times T}} \underbrace{- \sum_{s \in \mathcal{S}} \sum_{t \in \mathcal{T}} p(t|s) \log p(t|s) \sum_{d=1}^{D} w_d \cdot \tilde{p}(s,d)}_{H(p(t|s))}$$

$$\text{s. t. } \sum_{d=1}^{D} w_d \sum_{s \in \mathcal{S}} \sum_{t \in \mathcal{T}} f_i(s,t) \tilde{p}(s,d) \Big( p(t|s) - \tilde{p}(t|s,d) \Big) = 0 \tag{2}$$

$$\sum_{t \in \mathcal{T}} p(t|s) - 1 = 0, \quad \forall s \in \mathcal{S}, \qquad \sum_{d=1}^{D} w_d = M, \ w_d \geq 0, \ \forall d \in \mathcal{D}, \quad w_d \leq 1$$

We define the optimization problem as shown in equation 2. Here the objective function maximizes the model entropy $H(p(t|s))$ while minimizing the demonstrations weights $w$. The first constraint is the FEM between the empirical expected feature values and the generated feature value after the modification with the wight parameter. The second constraint is to make sure we generate a correct probability distribution of the sub-task prediction $p(t|s)$. The last constraint is to ensure that there are at least $M$ optimal demonstration in the dataset. We use Lagrange multiplier approach to derive the dual problem – as shown in (3) – that we solve to to find the weights $w$. The weights assigned to incorrect demonstrations are significantly lower than the correct ones. A complete derivation and implementation details can be found in the appendix A.

$$\min_{w \in \mathbb{R}^D, \lambda \in \mathbb{R}^N} \Lambda(\lambda, w) \equiv -\frac{1}{M} \sum_{d=1}^{D} w_d \Big( - \sum_{s \in \mathcal{S}} \tilde{p}(s,d) \log z_\lambda(s) + \sum_{i=1}^{N} \lambda_i \sum_{s \in \mathcal{S}} \sum_{t \in \mathcal{T}} \tilde{p}(t|s,d) f(s,t) \Big)$$

$$\text{s. t. } \sum_{d=1}^{D} w_d = M, \ w_d \geq 0 \ \forall d \in \mathcal{D}, \ w_d \leq 1 \tag{3}$$

Here $z_\lambda(s) = \sum_{t \in \mathcal{T}} \exp \left( \sum_{i=1}^{N} \lambda_i f_i(s,t) \right)$ is a normalization constant, and $\lambda$ is the Lagrange multiplier. A full pseudo code for the approach and how to solve the optimization problem can be found in appendix D.

## 4 Experiment and Results

We conducted experiments to investigate the following: (1) can a video segmentation algorithm be used to detect incorrect video demonstrations? (Section 4.1) (2) how well the weights generated by the proposed framework can differential among optimal, sub-optimal and incorrect demonstrations (Sections 4.2 and 4.3), and (3) what generates a better policy?: learning while leveraging incorrect demonstrations vs discarding incorrect demonstrations from the training pool (Section 4.4). We used two video datasets for experiments: our custom-designed imitation learning dataset *Tea-Making with a YuMi Robot* and publicly available activity recognition dataset *50-salads* [22].

### 4.1 Video Segmentation Vs Incorrect Demonstration Detection

Since the proposed framework used a video segmentation algorithm which performed sub-task segmentation and thereby generated a sequence of sub-task for each video, a natural confusion may arise: can we use a clustering algorithm to analyze the sequence of sub-tasks from a demonstration pool and identify the incorrect demonstrations? We performed a series of experiments to resolve this. All video segmentation algorithms make non-zero segmentation errors and the consequence of inaccurate segmentation is detrimental in identifying incorrect demonstrations. For example, the highest reported per-frame accuracy of

| Accuracy | Tolerance |
|----------|-----------|
| 0% | 5% |
| 30% | 10% |
| 50% | 20% |
| 70% | 30% |

Table 1: Sub-task accuracy

video-segmentation algorithm is $83\%$ on the publicly available 50-salads dataset [38]. The sub-task segmentation accuracy – which is relevant for detecting incorrect demonstration – reduces to $70\%$ with $30\%$ tolerance in per-frame accuracy (i.e. we allow $30\%$ of the frames from a sub-task to be incorrectly classified). This accuracy decreases significantly with reduced tolerance in per-frame accuracy. Table 1 summarizes the results. As an example, Fig. 2 shows the segmentation of a video from the *50-salad* dataset into 17 sub-tasks (each sub-task coded with different colors) using the best-performing segmentation algorithm reported in [38]. The algorithm has generated more than four sub-tasks with false labels. There is no known way to directly use such an output sequence to distinguish between correct vs incorrect demonstration.

### 4.2 Analysis of Demonstration Weights: Tea-Making with YuMi Robot

#### 4.2.1 The dataset

This custom-designed dataset has 36 videos showing three different human participants making a cup of tea – each participant performing 12 trials. The dataset was created through an IRB-approved user study. There are seven ground-truth sub-tasks in tea-making: *Turn on the oven, Add water, Add sugar, Add milk, Add tea bag, Stir, Turn off the oven*. Participants were asked to make tea while following any sequence of their choice but they must use both ingredients (i.e. milk and sugar). In majority of the resultant videos participants observed the following sequence: *Turn on the oven, Add tea bag, Add sugar, Add milk, Turn off the oven, Add water, Stir*. We consider these videos as **optimal demonstrations** and use the term **Seq_1** to denote them. Only a subset of videos show participants using the following sequence: *Turn on the oven, Add sugar, Add milk, Add tea bag, Turn off oven, Add water, Stir*. We consider these videos as **sub-optimal demonstrations** and use the term **Seq_2** to denote them. In a small subset of videos, the participants – based on our instructions – skipped one or two of the three critical sub-tasks namely, *Add tea-bag, Add hot water, Turn off the oven*. These videos are considered as **incorrect demonstrations** since they show a wrong or unsafe way of performing the tea-making task. We use the term **Seq_3** to denote them. Finally, to facilitate IL policy learning by real robots – which requires the joint angles of the robot for training – another 12 demonstrations were added to the dataset where the YuMi robot, instead of a human demonstrator, is remotely controlled to do the same tea-making task while following the same sub-task sequence as human demonstrators. All 48 videos are used for training and evaluations. A set of representative frames from three different types of video demonstrations are shown in appendix C

#### 4.2.2 Sub-task segmentation and feature generation

Using MS-TCN++ [38] we segmented the videos into $q = 7$ sub-task segments with a per-frame accuracy of 80%. Fig. 3 shows the segmentation of a video from the *tea-making* dataset . From each segment, 64 frames were uniformly sampled. Frames were reduced to $224 \times 224$ pixels and went through background subtraction before being fed to the feature generator (Section 3.2). The dimension of the generated features was $64 \times 32$ which was further reduced to $1 \times 32$ through a *max* function over the 64 frames. Appendix B shows a visual representation of some of the features generated by the model.

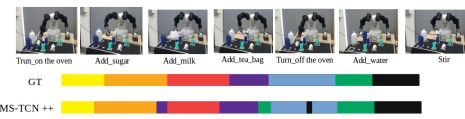

Figure 3: Top row: Sample frames from *tea-making* task representing the different sub-tasks. Middle row: The ground truth sub-task segmentation (each color represents a sub-task). Bottom row: sub-task segmentation by MS-TCN++ [38]. Incorrectly added sub-tasks are in purple, green, and black).

#### 4.2.3 Demonstration weights

The proposed framework, through equation (3), generates weights for different videos. This section discusses how these weights are used to identify optimal, sub-optimal, and incorrect demonstrations. For this, we create two demonstration sets: the first one has 6 optimal videos (**seq_1**: $d1$ *to* $d6$), 4 sub-optimal (**seq_2**: $d7$ *to* $d10$) and 2 incorrect ones ($d11$ *to* $d\_12$) while the second set has 8 optimal (**seq_1**: $d1$ *to* $d8$) and 4 incorrect videos ($d9$ *to* $d12$). Fig. 4 shows the weights for different demonstrations for these two sets after normalizing them to have values between $[0, 1]$. The weights are distributed in such a way that it is trivial to pick up two thresholds to differentiate optimal, sub-optimal, and

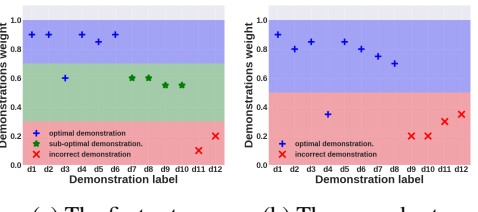

(a) The first set.  (b) The second set.

Figure 4: Weights for two sets of videos from the *Tea-Making with a YuMi Robot* dataset. It is trivial to pick up two thresholds to differentiate optimal ('blue'), sub-optimal ('green'), and incorrect videos ('red').

incorrect videos. For example, we used k-means algorithm [43] to cluster the weights while choosing the middle point between the cluster means as the threshold. The number of clusters is preset to 2 or 3, depending on the number of ground truth clusters. We experimented with two other clustering algorithms – Gaussian mixture model [43] and Mini-Batch K-Means [44] – both of which generated the same results. Detailed results are reported in appendix E.

Note, there is one false positive (FP) in each of these two sets, $d3$ and $d4$. Further analysis on these two FP videos indicates two possible sources of errors: 1) the solver that solves the non-convex,

non-linear optimization problem in (3) did not converge to the global minimum or (2) the feature generator (Section 3.2) failed to identify a good set of features. Section 4.2.4 discusses the implication of generating erroneous weights.

### 4.2.4 Robustness of performance

We investigated the robustness of the proposed method in detecting sub-optimal and incorrect demonstrations. For this, we made a demonstration pool with 10 optimal demonstrations (**seq_1**) and one sub-optimal or incorrect demonstration (**seq_2** or **seq_3**). Keeping the number of optimal demonstrations fixed at 10, we increased the incorrect (or sub-optimal) demonstrations in the dataset until they become $44\%$ of the dataset (i.e., 10 optimal and 8 incorrect or sub-optimal). Tables 2 and 3 show accuracy, recall, and precision in different cases. Here, a false positive (FP) indicates an optimal video demonstration inaccurately detected as incorrect (or sub-optimal) while a false negative (FN) is an incorrect (or sub-optimal) video incorrectly identified as optimal. True positive (TP) and true negative (TN) are also defined accordingly.

The proposed framework can detect sub-optimal and incorrect video demonstrations with an average accuracy of $86\%$ and $90\%$, respectively. However, when detecting incorrect demonstration, the metric that carries most significance is Recall ($\frac{TP}{TP+FN}$). The proposed framework shows $100\%$ recall, both for **Seq_2** and **Seq_3**. In other words, the framework can correctly identify every incorrect/sub-optimal demonstration in the dataset. The implication of this is significant: the framework can pass to the policy learning algorithm a training set consisting of purely optimal demonstrations and thereby learning a highly accurate policy [15, 16, 20, 45] (more discussion in Section 4.4). While the framework generates some false positive results (as shown by Precision values $78\%$ and $83\%$, respectively ), they do not directly jeopardize the quality of the learned policy. However, too many false positive may cause many demonstrations to be thrown away, resulting in a rapid shrinking of the demonstration pool which is undesirable for realistic applications where collecting demonstration is costly.

| #Seq_1 | #Seq_2 | Accuracy | Recall | Precision |
|---|---|---|---|---|
| 10 | 1 | 90 | 100 | 67 |
| 10 | 2 | 92 | 100 | 80 |
| 10 | 4 | 85 | 100 | 80 |
| 10 | 6 | 81 | 100 | 80 |
| 10 | 8 | 83 | 100 | 84 |
| Average | | $86\%$ | 100 | 78 |

Table 2: Robustness with Seq_2

We performed one other experiment where we used video segmentation algorithms with various accuracy and investigated the robustness of performance of the proposed framework. The detection accuracy of the proposed framework remains the same even with a video segmentation accuracy of $100\%$ (achieved through manual segmentation). Results from this experiment are in appendix E.

## 4.3 Analysis of Demonstration Weights: The 50-salads

### 4.3.1 The dataset

The *50-Salads* is a public dataset containing 50 videos of 25 actors preparing two different types of salads. The videos are recorded from an overhead camera and the average video duration is 6.4 minutes. There are 17 unique sub-tasks involved with preparing the dressing and mixing different ingredients (Cheese, cucumber, lettuce, tomato, dressing). However, the average num-

| #Seq_1 | #Seq_3 | Accuracy | Recall | Precision |
|---|---|---|---|---|
| 10 | 1 | 90 | 100 | 67 |
| 10 | 2 | 92 | 100 | 80 |
| 10 | 4 | 93 | 100 | 89 |
| 10 | 6 | 88 | 100 | 86 |
| 10 | 8 | 89 | 100 | 89 |
| Average | | $90\%$ | 100 | 83 |

Table 3: Robustness with Seq_3

ber of sub-tasks in different videos is 20 as different actors follow different sub-task sequences to prepare the salad. Since it is not a custom-designed datasets, we had to devise a strategy to create incorrect demonstrations. Being an activity recognition dataset, *50-salads* is a balanced dataset where half of the videos starts with sub-tasks related to *preparing dressing* and the other half with that related to *cutting and mixing different ingredients*. We used the first half of the dataset to create the incorrect demonstrations by removing all frames related to *preparing dressing*. Thus the **optimal demonstrations** are those videos where actors *cut and mix ingredients* and then *prepare the dressing* while **incorrect demonstrations** are the ones where actors miss the sub-tasks related to *preparing dressing*. Because of the limitation of the dataset, we focused only on detecting the incorrect demonstration, not the **sub-optimal** demonstrations. Fig. 2 shows example frames from the *50-salads* dataset.

### 4.3.2 Sub-task segmentation and feature generation

Using MS-TCN++ [38] we segmented each videos into $q = 17$ sub-task segments with a per-frame accuracy of $77\%$ which is very close to the accuracy reported in the original paper [38]. Fig. 2 shows the segmentation of a video from the dataset. Feature generation followed the same strategy we adopted for the *Tea-making with YuMi Robot* dataset.

### 4.3.3 Demonstration weights

To analyze the weights of optimal vs incorrect demonstrations we created a demonstration set with 10 correct demonstrations ($d1$ to $d10$) and 2 incorrect ones $d11$ to $d12$. Fig. 5 shows the normalized weights for different demonstrations.

### 4.3.4 Robustness of performance

We evaluated the robustness of the proposed method in detecting only the incorrect demonstrations in the *50-Salads* dataset. We made a demonstration pool with 10 optimal demonstrations and 1 incorrect one, and gradually increased the number of incorrect demonstrations in the pool. Table 4 shows accuracy, precision, and recall for different settings. From Table 4, the average accuracy is $85\%$ which is lower than that with the *Tea-making with YuMi Robot* dataset. This is because *Salad-preparation* is much more complex task than *Tea-making* with more than a double

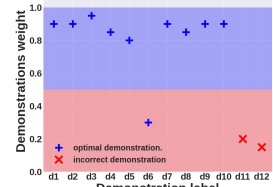

Figure 5: Weights for the *50-Salads* dataset.

number of sub-tasks. However, the recall performance remains $100\%$ – which means we never missed an incorrect demonstration ($FN = 0$) – a desirable property for eventually learning an accurate policy. However, unlike the *Tea-making with YuMi Robot* dataset, we could not empirically investigate the policy accuracy for this dataset due to lack of robot joint angle data.

### 4.4 Leveraging Vs Discarding Incorrect Demonstrations: Which Gives the Better Policy?

For the tea-making task, we created a demonstration set containing varying number of incorrect demonstrations and investigated the effect of having incorrect demonstrations on the accuracy of the learned policy. We studied three different strategies for policy learning: i) we use the proposed framework to detect incorrect demonstrations and discard them from the training set before learning a policy using BC/BC-RNN [46, 2], ii) we use BC/BC-RNN to learn a policy without discarding the incorrect demonstrations iii) we use two

| # Seq_1 | # Seq_3 | Accuracy | Recall | Precision |
|---------|---------|----------|--------|-----------|
| 10 | 1 | 90 | 100 | 67 |
| 10 | 2 | 83 | 100 | 67 |
| 10 | 4 | 85 | 100 | 80 |
| 10 | 6 | 81 | 100 | 77 |
| Average | | $85\%$ | **100** | **73** |

Table 4: Robustness with Seq_3 for the *50-Salads*

contemporary IL approaches – DEMO-DICE [28] and R-MaxEnt [25, 26] – that leverage incorrect demonstrations for policy learning but assign poor weights to them in the learning process. Fig. 6 shows the policy accuracy in different settings where policy-accuracy is defined as *(Number of correctly chosen actions / Total number of executed actions until the goal is reached)* $\times 100$. When incorrect demonstrations are present in the training set, Demo-Dice and R-MaxEnt learned a better policy than BC/BC-RNN by assigning poor weights to incorrect videos. However, BC/BC-RNN learned far better policy than Demo-Dice and R-MaxEnt when the incorrect videos were removed by the proposed framework based on their poor weights. Implementation details and code links for the policy learning algorithms can be found in appendix D.

## 5 Conclusion

We presented a novel approach for detecting incorrect video demonstrations of any multi-stage sequential task. The proposed method leverages entropy analysis to identify video(s) in the demonstration pool that follow a 'disruptive' task-sequence and assign poor weights to them. Using two different datasets we demonstrated the effectiveness of our approach. The proposed framework can be used as a pre-processing step for any visual IL algorithm to improve the accuracy of the learned policy.

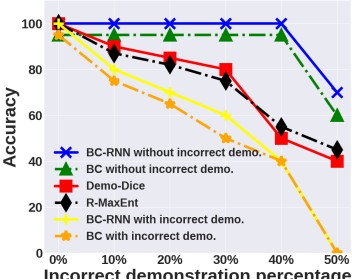

Figure 6: Accuracy of the tea-making task policy learning.

## 6 Limitations and Future Work

The proposed framework requires a value for $q$ (the number of sub-tasks) at the beginning of the pipeline. Techniques for sub-goal identification from video – an active research area in computer vision – can be employed for automatically generating a value for $q$. The proposed work is also limited in detecting missing sub-tasks without providing any information about the importance of these sub-tasks. Introducing risk parameter for each sub-tasks and learning those parameters from the dataset is a possible solution.

## 7  Acknowledgement

This work was supported in part by the National Science Foundation (IIS 1830597).

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
