# OpenReview forum: "Detecting Incorrect Visual Demonstrations for Improved Policy Learning"
_robot-learning.org/CoRL/2022/Conference — CoRL 2022 Poster_

### Official Review · Reviewer_key5 · 2022-07-28

**Originality:** Good
**Technical Quality:** Good
**Clarity Of Presentation:** Good
**Impact:** 3

**Recommendation:**

Weak Accept: I recommend accepting the paper, but will not argue for my recommendation if the majority of other reviewers have a different opinion.

**Summary:**

The quality of demonstration data is crucial for imitation learning. This paper proposes a method to detect and filter out erroneous data from visual demonstrations (i.e., video streams) of sequential tasks, such as tea making or salad preparation. The proposed method first uses off the shelf methods to segment out subtasks from videos and then uses prior work to extract subtask-relevant features. Then, using the set of task demonstrations, it seeks to learn a predictive model of the next subtask given the current state and, in this process, identify which video demonstrations are erroneous (and therefore should be downweighted). The paper modifies the classic maxEntIRL optimization by posing a mini-max entropy optimization: find a weight for each demonstration that *minimizes* entropy of the sub-task predictive model (i.e., choose demonstrations that enables confident subtask prediction) while finding the distribution that *maximizes* entropy over the predictive model.

**Issues:**

Section 3:

–Please clarify the technical correctness of equation (3) – the dual problem to the mini-max game. As stated above in the weaknesses, I believe that the dual problem should be:
$$
(w^*, p^*(t|s), \lambda^*, \nu^*) = \arg \min_{\lambda, \nu} \arg \min_{w} \max_{p(t|s)} \mathcal{L}(w, p(t|s), \lambda, \nu)
$$
where $\mathcal{L}$ is the Lagrangian which includes the entropy objective plus $\lambda^\top \text{(feature expectation matching constraint)}$ plus $\nu \text{(valid probability distribution constraint)}$. It would then be helpful to state what are the first order necessary conditions for optimality, for $w^*$, $p^*(t|s)$, $\lambda^*$, and $\nu^*$, and comment on how the solver works and if there is any concern about obtaining an approximate solution.

Section 4:

–Table 1 is unclear on its own. For example, what do the rows of the table correspond to? Please clarify this.

Section 4.2.3:

–”The weights are distributed in a way that it is trivial to pick up two thresholds to differentiate optimal…” This is a strong claim. For example, the thresholds chosen in Figure 4 do not achieve a 100% accuracy in filtering out the optimal demonstrations vs. incorrect demonstrations vs. suboptimal demonstrations. Discussing a principled way to go about choosing a threshold, and what happens when the threshold incorrectly characterizes an optimal demonstration as incorrect, etc. would strengthen this claim. This same concern exists for Section 4.3.3 where the same statement is reiterated.

Section 4.4:

–”Fig 6 shows policy accuracy …When incorrect demonstrations are present in the training set….poor weights to incorrect videos.” The definition of policy accuracy relies on the number of correct actions chosen – where does the notion of a correct action come from?


**Quality Of The Limitations Section:**

Limitations are addressed clearly

**Reviewer Expertise:**

4: The reviewer is confident but not absolutely certain that the evaluation is correct

**Robotics Focus:**

Sufficient demonstration on hardware

**Strengths And Weaknesses:**

Strengths:
*************
–Interesting problem of identifying erroneous / low-quality demonstration data.

–Valuable direction investigating erroneous high-dimensional data, like images.

–Interesting approach of posing this mini-max optimization to determine which demonstrations are helpful vs. erroneous for predicting subtasks.

--Interesting comparison of leveraging vs. discarding incorrect demonstrations when it comes to policy learning.

Weaknesses:
*************
–The paper often makes claims without substantiating them properly with citations or evidence. The specifics of these instances are detailed below on a section-per-section basis.

–The intuition behind why this mini-max optimization procedure should yield “good” weights that detect erroneous visual data could be strengthened.

–The technical correctness of equation (3) – the dual problem to the mini-max game – seems questionable. For example, the maximization over the distribution P disappears in the dual problem. Also, the explanation for how this optimization is solved is unclear; solving mini-max games is not trivial in general, and it seems like the proposed method is solved in a way that will likely be an approximate (and not exact) solution to the non-convex game. Providing a more detailed derivation of the dual problem, and discussing the solution quality would strengthen the paper. For example, I believe that the dual problem is:
$$
(w^*, p^*(t|s), \lambda^*, \nu^*) = \arg \min_{\lambda, \nu} \arg \min_{w} \max_{p(t|s)} \mathcal{L}(w, p(t|s), \lambda, \nu)
$$
where $\mathcal{L}$ is the Lagrangian which includes the entropy objective plus $\lambda^\top \text{(feature expectation matching constraint)}$ plus $\nu \text{(valid probability distribution constraint)}$. It would then be nice to state what are the first order necessary conditions for optimality, for $w^*$, $p^*(t|s)$, $\lambda^*$, and $\nu^*$.

Intro:

–Nitpick: “Visual imitation learning …. Is probably the most feasible…” This is quite a strong claim. I’d suggest soften it by saying that “Visual imitation learning is a promising way for …”

–“The state of the art…-almost all of which involve non-visual demonstrations–[is learning] weights…” Instead of [is learning] it should be [learn]. Also, it could be helpful to include a few citations of which state of the art approaches are being referred to here.

–”...2) individual steps of a multi-step task - hereafter referred [to] as sub-tasks…” missing the [to] in the sentence.

–”We bridge this gap…. Which does not have any restrictive assumptions.” Instead of making the broad (and somewhat unsubstantiated claim) that there are no restrictive assumptions in the proposed method, it would be good to focus on the explicit differences between the prior work and the proposed work in this context – for example, mentioning explicitly that the proposed method does not require expert labeling or a simulator for data augmentation.

Section 2:

–”...where a single incorrect action may or may not cause the algorithm to eventually learn a completely incorrect policy….however, a single incorrect action can lead to an incorrect policy” This statement is not well-explained or well-substantiated. It seems as though regardless of if the task is multi-step or not, a single incorrect action can lead to either catastrophic failure or simply a marginally suboptimal behavior.

Section 3.1:

–”Here, q is a hyperparameter which does not require fine-tuning as we do not need very accurate sub-task segmentation.” Accurately identifying subtasks (and, related, the number of sub-tasks) seems like an essential prerequisite for the proposed method to work properly. This claim should either be removed or substantiated through some sensitivity analysis.

–”The proposed framework is tolerant to typical errors in the existing video segmentation algorithms.” This claim should either be removed, or substantiated through sensitivity analysis / quantitative results.

Section 3.3:

–It would be helpful to provide intuition behind what the value of the weights means. If the weight is = 0, does this mean that the demonstration is not good and therefore is not used, and if the weight = 1, then it is an “optimal” demonstration and should be used to the fullest extent?


**Summary Of Recommendation:**

Although the problem addressed in this work is interesting, I recommended weak reject on the basis of the mathematical formulation of the core method seeming incorrect/incomplete (i.e., equation 3) thereby bringing into question the technical correctness of the proposed method and results, and the need to further empirically / theoretically substantiate some of the strong claims made in this paper (e.g., the "triviality" of picking thresholds of correct and incorrect demonstrations, sensitivity having accurate sub-task segmentation).

--------------------------------------

After discussion, I have changed my recommendation to "weak accept"

---

> ### Author Response · Authors · 2022-08-20
> **Response to reviewer key5**
>
> Thank you very much for your insightful comments. We will address the concerns raised under weakness. We also want to mention that we presented experiments with a real robot (with respect to the assessment. Robotics Focus: Highly relevant to robotics but no hardware experiments)
>
> Nitpicks: Many of your suggestions will actually improve the readability of the paper and we have incorporated them. Thank you! The ones related to technical correctness, sensitivity analysis, and threshold choice are clarified below.
>
>     Concern #1: Technical correctness of equation (3)
>
> Clarification: The dual problem in equation (3) is derived using Lagrange multiplier approach. Although our original manuscript -- due to limited space -- did not include the step-by-step derivation of the dual problem,   we made sure to include that in the appendix. We strongly believe that the derivation – as it appears in the appendix – which has led us to equation (3) is correct. This derivation, however, cannot lead to the dual problem that you mentioned. If you could kindly direct us to potential errors in our derivation, we would be happy to work on it.  A clarification can be added here, In the primal problem we solve for $p^{\ast}{(t|s)}$ and $w^{\ast}$, once we derive the dual problem we will solve for $w^{\ast}$ and  $\lambda^{\ast}$ the optimal values for p^{\ast}(t|s) can be generated from equation 9 in the appendix given the current features f(s,t), but we do not need these values as we only looking for the weights $w^{\ast}$.
>
>     Concern # 2: The explanation for how this optimization is solved is unclear
>
> Clarification: We acknowledge that this implementation detail is missing in the original manuscript. We add the following missing information: we solve the non-convex quadratic problem in equation (3) using the Sequential Quadratic Programming (SQP) algorithm (Implementation: {https://www.mathworks.com/help/optim/ug/constrained-nonlinear-optimization-algorithms.html\#bsgppl4}); see, for example, Chapter 18 of Nocedal and Wright [1]. The SQP algorithms generalize Newton's method for constrained optimization problems. In each iteration, the Hessian of the Lagrangian function is approximated in a quasi-Newton style. The algorithm then solves the resulting quadratic program and finds the next iteration using the line search procedure. This algorithm may not converge to the global minimum, but our experiments find it to perform very well.
>
>     Concern # 3. The intuition behind why this mini-max optimization procedure should yield “good” weights that detect erroneous visual data could be strengthened.
>
> Clarification: We acknowledge that our assumption that section 3.3 adequately describes the intuition behind using the min-max optimization may not be true. Therefore, after the following sentence in the current introduction -- “Accordingly, the intuition behind the proposed framework is to identify groups of spatio-temporal features as the representatives of different sub-tasks (Sections 3.1 and 3.2) and analyze the internal consistency of these features over all videos in the demonstration pool (Section 3.3) -- we will add the following explanation -- “To achieve that we propose to solve a minmax optimization problem where we choose the model with the maximum entropy (the most uniform) and at the same time minimizing the demonstrations weight so we can achieve a saddle point with the minimum number of demonstrations.” We are open to further comments on this.
>
>
>
>     Concern #4. Table 1 is unclear on its own. For example, what do the rows of the table correspond to? Please clarify this.
>
> Clarification: Section 4.1 provides explanation for Table 1: How well can we infer if a video is incorrect/correct only by looking at the result of a video segmentation algorithm? For example, if a video segmentation algorithm produces a correct sequence of sub-tasks from any video in a highly reliable manner, we can simply infer a video to be incorrect if its sub-task sequence does not match with those from the training data. Table 1 is to empirically show that this strategy is not feasible while considering 50-salad dataset as an example. Video segmentation algorithms produce erroneous sub-task sequences, making it non-trivial to infer if a video is correct/incorrect. Accordingly, the rows in Table 1 show the sub-task segmentation accuracy of the top-most video segmentation algorithm (for the 50-salad) for different tolerance levels; here tolerance indicates the percentage of frames from a sub-task segment that we allow to be incorrect.  For example, the first row indicates that if the tolerance is 5% (i.e., if a sub-task consists of 20 frames, we will consider a segmentation as correct if 1 out of these 20 frames is mis-classified as other sub-task, all based on ground truth data), the accuracy of the algorithm drops down to 0%. Here, the accuracy means how well the resulting sequence matches the ground truth.

---

> > ### Author Response · Authors · 2022-08-20
> > **Response to reviewer Key5 (Continue)**
> >
> >     Concern#5. Choice of threshold (section 4.2.3)
> >
> > Clarification: We did not report extensive experiments on choosing the best threshold and acknowledge that it will improve the quality of the paper. Currently we are using a k-means algorithm to cluster the weights and the threshold is chosen as the middle point between the resulting means. The number of clusters is preset to 2 in case of correct/incorrect demonstrations. We will include a more thorough analysis in the camera ready version. However, the original manuscript (lines #299-303) offers explanations on the comment regarding “....incorrectly characterizes an optimal demonstration as incorrect”. We already report that this is an example of false positive and such a video will be thrown away by the framework. With respect to policy learning, it does not jeopardize the quality of the learned policy. However, too many false positives may cause many demonstrations to be thrown away, resulting in a rapid shrinking of the demonstration pool which is undesirable for realistic applications where collecting demonstrations is costly.
> >
> >     Concern# 6. Fig 6 shows policy accuracy …When incorrect demonstrations are present in the training set….poor weights to incorrect videos.” The definition of policy accuracy relies on the number of correct actions chosen – where does the notion of a correct action come from?
> >
> > Clarification: The notion of correct action comes from the ground truth. In order to report policy accuracy as a performance metric, we manually labeled the correct action for each segmented sub-task.
> >
> >     Concern # 7.  Here, q is a hyperparameter which does not require fine-tuning as we do not need very accurate sub-task segmentation.” Accurately identifying subtasks (and, related, the number of sub-tasks) seems like an essential prerequisite for the proposed method to work properly. This claim should either be removed or substantiated through some sensitivity analysis.
> >
> > Clarification: We are confident about our comment on q (please check our response for Concern #1 of reviewer K64Q) yet acknowledge that a sensitivity analysis is needed. However, we may not be able to complete such an analysis before the review window closes. Therefore, we will remove this claim.
> >
> >      Concern # 8.  The proposed framework is tolerant to typical errors in the existing video segmentation algorithms.” This claim should either be removed or substantiated through sensitivity analysis / quantitative results.
> >
> > Clarification: This claim is based on the 83% accuracy of the top-most video segmentation algorithm for the 50-salad dataset. Therefore, instead of a generic claim we modified the sentence to match this result: “The proposed framework is tolerant to typical errors in the existing video segmentation algorithms up to 83% accuracy”
> >
> >     Concern # 9. It would be helpful to provide intuition behind what the value of the weights means. If the weight is = 0, does this mean that the demonstration is not good and therefore is not used, and if the weight = 1, then it is an “optimal” demonstration and should be used to the fullest extent?
> >
> > Clarification: The weights are normalized between [0,1], 0 means the demonstration is incorrect based on the analysis of the model entropy and 1 means the demonstration is complying with most of the demonstration set which we infer as a correct demonstration.
> >
> > [1] Jorge Nocedal and Stephen Wright. Numerical optimization. Springer Science & Business Media, 2006.

---

### Official Review · Reviewer_JG5k · 2022-08-01

**Originality:** Very Good
**Technical Quality:** Very Good
**Clarity Of Presentation:** Very Good
**Impact:** 4

**Recommendation:**

Weak Accept: I recommend accepting the paper, but will not argue for my recommendation if the majority of other reviewers have a different opinion.

**Summary:**

This paper attempts to automatically detect incorrect video demonstrations for multi-step sequential tasks. The method learns a stochastic model of subtask sequence and generate weight for demonstrations using feature expectation matching with principle of maximum entropy. The approaches are verified with two datasets.

**Issues:**

1.	As mentioned in the “weaknesses” part, what are some situations where human cannot directly distinguish their incorrect behaviors or must spend lot of human efforts for doing that, so that detecting using the algorithm is necessary?

2.	For sequential tasks, what if most parts of the demonstration are correct and only a part of it is incorrect? Could the work be adapted to address this by labelling incorrect for subtasks so that this helps keeping important information?

3.	 Line 153 is a little bit unclear. Is it for each video segment or each video demonstration, that a set of ranked features is extracted?

Minor issues:

1.	Line 97 The ‘proposed’ framework

2.	Line 235 Seq_2 is not fully bolded.


**Quality Of The Limitations Section:**

Limitations are addressed clearly

**Reviewer Expertise:**

4: The reviewer is confident but not absolutely certain that the evaluation is correct

**Robotics Focus:**

Sufficient demonstration on hardware

**Strengths And Weaknesses:**

Strengths:

•	The paper is clearly written in detail.

•	The authors provide a comprehensive survey over related work. The author also pinpoints the difference between the existing work and their new approach.

•	The experiment analysis of no false negative detection is convincing.

Weaknesses:

•	The motivation emphasizes on detect visual incorrect demonstrations instead of suboptimal demonstrations. And in the statement the incorrectness often come from obvious missing steps or wrong step orders. One concern is that for such obvious mistake, it is very easy to have the demonstrator label it as incorrect or failure or directly delete the demonstration if they conduct a dangerous/incorrect behavior.


**Summary Of Recommendation:**

The paper proposes to solve the problem of detecting incorrect visual demonstrations for multi-step sequential tasks. The paper is well-written and overall technical sound. The literature review is thorough and experiment results are convincing. The only problem is the necessity of leveraging the algorithm if human labelling is very easy.

---

> ### Author Response · Authors · 2022-08-21
> **Response to reviewer JG5k**
>
> Thank you very much for your thoughtful comments. We will clarify your concerns listed under Weakness and Issues
>
>     Concern #1. As mentioned in the “weaknesses” part, what are some situations where human cannot directly distinguish their incorrect behaviors or must spend lot of human efforts for doing that, so that detecting using the algorithm is necessary?
>
> Clarification: We believe as a research community we are working toward the overarching goal of having LfD-powered robots in the wild where the robot can learn from grandma how to make a cup of tea and etc. While it is easy for a robot developer to notice and discard an incorrect demonstration from the training pool, it is not easy for grandma – who does not know how a demonstration can possibly be incorrect for a complex learning algorithm – to tell the algorithm to discard that demonstration. Even if grandma intuitively understands that a demo she gave is wrong, we have to build a wrapper on the learning algorithm to allow her to communicate that information to the algorithm. We believe this is related to the problem of bi-directional algorithmic transparency, a contemporary topic that we are not exploring in this paper. Instead, we are developing an automated framework which will take all video demonstrations that grandma makes and identify the incorrect ones before sending them to the policy learner.
>
>     Concern #2. For sequential tasks, what if most parts of the demonstration are correct and only a part of it is incorrect? Could the work be adapted to address this by labelling incorrect for subtasks so that this helps keep important information?
>
> Clarification: Thank you for pointing to this interesting topic. Some existing works (non-visual) have used failed demonstrations, to various degrees, in the learning process [1]. However, the challenge of doing this for visual imitation learning is identifying which sub-task segment(s) in a video is incorrect. In the context of our work, it is about identifying what causes a particular video to have a higher entropy than others, instead of only detecting that a video has unusually high entropy (which is indirectly done by the min-max problem in the proposed framework).  This requires a more intelligent perception framework along with a slightly different optimization problem. This is the topic of our ongoing investigation.
>
>     Concern #3. Line 153 is a little bit unclear. Is it for each video segment or each video demonstration that a set of ranked features is extracted?
>
> Clarification: It will be for each video segment (each subtask), it can be understood as a 1 vector of features as a latent representation for each segment.
>
>     Minor issues
> All minor issues you identified are fixed and will be uploaded for the camera ready version. Thank you.
>
>
> [1] K. Shiarlis, J. Messias, and S. Whiteson. Inverse reinforcement learning from failure. 2016.

---

### Official Review · Reviewer_av9b · 2022-08-01

**Originality:** Very Good
**Technical Quality:** Very Good
**Clarity Of Presentation:** Very Good
**Impact:** 4

**Recommendation:**

Weak Accept: I recommend accepting the paper, but will not argue for my recommendation if the majority of other reviewers have a different opinion.

**Summary:**

This paper addresses an important problem in learning from video demonstrations, which is the presence of incorrect or suboptimal demonstrations in the dataset. Given a raw video input, it first performs sub-task segmentation on the video, and then use a feature generator to extract task-relevant features. Then it is put into an optimization problem solver to learn the demonstration weights, and eventually weight the demonstrations differently according to their demonstration quality. It is able to surpass the performance of IL models in the setting of imperfect demonstrations.

**Issues:**

As clarifications:

Regarding the dataset labeling:
- Does the dataset always come with ground truth labels of incorrect demonstrations? What happens when such labels are not available?

Regarding the feature generator:
- The feature generator extracts relevant task features from the videos. I'm curious whether this requires any assumption on the video demonstration type. The 50 salads dataset seems to have clear visual distinctions between different tasks. What if we are dealing with some robot manipulation tasks that does not involve large difference in visual scenes? How is the performance of this feature generator on common real robot tasks?


**Quality Of The Limitations Section:**

Additional details required

**Reviewer Expertise:**

3: The reviewer is fairly confident that the evaluation is correct

**Robotics Focus:**

Sufficient demonstration on hardware

**Strengths And Weaknesses:**

Strengths:

- Important problem setting, intuitive motivation of learning different weights for different quality samples

- Interesting discussion on "Leveraging Vs Discarding Incorrect Demonstrations" - this mirrors the discussion between offline RL and BC on their performance on different datasets. It makes sense that this methods performs better than BC / BC-RNN for datasets with incorrect demonstrations, since it performs the filtering function that filters out suboptimal trajectories.

- clear presentation of the overview figure about model pipeline and model architecture. nice visualization of segmentation results and how they correspond to the video frames

- convincing results demonstrated on real, multi-step sequential tasks

Weaknesses:

- For Figure 6: For "BC-RNN without Incorrect Demo", are the number of *correct* demos the same with those in Demo-Dice? If so, I would expect Demo-Dice have similar performance with BC / BC-RNN, because Demo-Dice should be able to eliminate the effect of the incorrect demos?

- For related work section on learning from suboptimal dataset, there should be discussions made to offline RL methods that learns from large dataset from different data quality. In particular, the weighted-BC line of work featuring advantage-weighted BC also learn weights for samples, and the weights from from advantage estimation. Such works include:

1. Critic Regularized Regression

    by Ziyu Wang, Alexander Novikov, Konrad Zolna, Jost Tobias Springenberg, Scott Reed, Bobak Shahriari, Noah Siegel, Josh Merel, Caglar Gulcehre, Nicolas Heess, Nando de Freitas

2. AWAC: Accelerating Online Reinforcement Learning with Offline Datasets

    by Ashvin Nair, Abhishek Gupta, Murtaza Dalal, Sergey Levine

3. Offline Reinforcement Learning with Implicit Q-Learning

    by Ilya Kostrikov, Ashvin Nair, Sergey Levine

**Summary Of Recommendation:**

I think the paper addresses an important problem in imitation learning and have explained the motivations well. It gives a novel approach of learning demonstration weights that is applicable in real-world robotics setting. More clarifications and references are needed but in general the paper have convincing results.

---

> ### Author Response · Authors · 2022-08-20
> **Response to reviewer av9b**
>
> Thank you very much for your thoughtful comments. We will clarify your concerns listed under Weakness and Issues.
>
>     Concern #1. For Figure 6: For "BC-RNN without Incorrect Demo", are the number of correct demos the same with those in Demo-Dice? If so, I would expect Demo-Dice have similar performance with BC / BC-RNN, because Demo-Dice should be able to eliminate the effect of the incorrect demos?
>
> Clarification: The number of correct demonstrations is the same for all approaches. We attribute the poor performance of Demo-Dice to its design. Demo-Dice is designed to detect demonstrations that are structurally correct but plagued with a limited amount of added noise. However, for this paper Demo-Dice was challenged with detecting demonstrations that are structurally incorrect (e.g., missed sub-tasks, wrong sequencing of sub-tasks).  In addition to that, to the best of our knowledge, Demo-Dice has never been tested on multi-step sequential tasks such as tea-making or salad preparation.
>
>     Concern #2. For related work section on learning from suboptimal dataset, there should be discussions made to offline RL methods that learns from large dataset from different data quality. In particular, the weighted-BC line of work featuring advantage-weighted BC also learn weights for samples, and the weights from advantage estimation. Such works include..
>
> Clarification: Thanks! Suggested references are now in the manuscript along with a discussion on them in the related work section.
>
>     Concern #3. Does the dataset always come with ground truth labels of incorrect demonstrations? What happens when such labels are not available?
>
> Clarification: The framework does not require the dataset to come with any labels. We labeled it only to calculate and report the accuracy of the proposed framework. The framework only requires that most videos in the training set (ideally, more than 50%) are correct (please see our response to reviewer K64Q.
>
> Concern#1b that elaborates on the validity of this assumption). It can then determine which videos have ‘feature-sequences’ that are substantially different than the majority of the other videos. Those videos are tagged as incorrect demonstrations.
>
>      Concern # 4. The feature generator extracts relevant task features from the videos. I'm curious whether this requires any assumption on the video demonstration type. The 50 salads dataset seems to have clear visual distinctions between different tasks. What if we are dealing with some robot manipulation tasks that does not involve large difference in visual scenes? How is the performance of this feature generator on common real robot tasks?
>
> Clarification: This framework and the feature generator are designed for multi-step sequential tasks. Typical manipulation tasks are one-step trajectory learning and this framework will need modifications to perform well in such tasks. This is an interesting direction that we are planning to explore next. Thank you for the question.

---

### Official Review · Reviewer_K64Q · 2022-08-02

**Originality:** Fair
**Technical Quality:** Good
**Clarity Of Presentation:** Good
**Impact:** 2

**Recommendation:**

Weak Reject: I recommend rejecting the paper, but will not argue for my recommendation if the majority of other reviewers have a different opinion.

**Summary:**

This paper introduces a framework for detecting incorrect visual demonstrations for imitation learning. The framework first performs temporal segmentation of sub-tasks. Then it uses a maximum entropy model to find the most consistent group of sub-tasks as the correct demonstrations. The experiments are performed on the 50-salad dataset and a customized dataset showing tea making by human.

**Issues:**

The author can respond to the "Weakness" part of the review.

**Quality Of The Limitations Section:**

Limitations are addressed clearly

**Reviewer Expertise:**

2: The reviewer is willing to defend the evaluation, but it is quite likely that the reviewer did not understand central parts of the paper

**Robotics Focus:**

Sufficient demonstration on hardware

**Strengths And Weaknesses:**

Strength:
- The method is well-motivated and points out an important problem for imitation learning.
- The method is backed up by theoretical results and analysis.
- The presentation of the paper is overall clear.

Weakness:
- The whole framework is based on video temporal segmentation. What if the segmentation model has really bad performance? How will that affect the final performance of incorrect demonstration detection? For example, since the proposed framework uses a maximum entropy model, if most (> 50% or > 70%) of the demonstrations are wrong or imperfect, can the framework still give desired results by finding the most consistent group of sub-tasks? Or the video segmentation model has a small but the same bias of error on all demonstrations?
- The experimented data are both small in scale as well as very limited in the variations of their scenarios, i.e. 36 videos of human making tea with fixed background and scene. How about demonstration videos in the wild, e.g. making tea in different kitchens by different people?
- Though the overall goal of this paper is to deal with learning from demonstration, the core contribution of the proposed approach seems like an optimization approach for matching or finding consensus that uses max entropy and has no learning components in it. It raises the alert about one of the desk-reject criteria of "No learning" on the CoRL 2022 website: https://corl2022.org/instructions-for-authors/.

**Summary Of Recommendation:**

Though the problem is important, it's not clear how well the proposed method can perform if the video segmentation model is wrong or unreliable. The experiments are also a little weak for supporting the claim made in the paper.

----

I thank the authors for their response. I will still keep my original "weak reject" score.

My main concerns still remain after the rebuttal. I'm not convinced that the video model presented in this paper is able to generalize to different scenes and scenarios, due to the small scale and limited diversity of the experimented data. This is irrelevant to whether the model takes raw pixels as input or uses a pre-trained feature extraction module. Based on my experience of working on video temporal segmentation and my understanding of the state-of-the-art performance on this problem, I don't believe the proposed entropy method can scale to video data with a larger scale and more diversity in the wild.

By definition of "weak reject", I "will not argue for my recommendation if the majority of other reviewers have a different opinion".

---

> ### Author Response · Authors · 2022-08-19
> **Response to reviewer K64Q**
>
> Thank you very much for your insightful comments. We will address the concerns raised under weakness.
>
>     Concern #1a: The whole framework is based on video temporal segmentation. What if the segmentation model has really bad performance? How will that affect the final performance of incorrect demonstration detection? Or the video segmentation model has a small but the same bias of error on all demonstrations.
>
> Clarification: Video segmentation – performed through any off-the-shelf algorithm – is one pre-processing step of the framework. In the subsequent step, spatio-temporal features are extracted from different video segments and are employed by the framework for decision making. Because of our powerful feature extractor, the framework performs well with any reasonably good video segmentation output. We have experimental data (which was not reported in the original manuscript) that shows the framework’s detection performance is 90% even when the video segmentation accuracy is 100% (achieved through manual segmentation). We will report another data point, on the camera ready version, to show that a video segmentation accuracy far less than 83% does not significantly affect the framework’s performance.  However, it is true that if a video segmentation algorithm has extremely poor (e.g., random-chance) accuracy, strong features won’t be enough to generate good performance. But we want to shed light on the fact that the recent development in computer vision research has enabled this whole paradigm of visual imitation learning. Video segmentation algorithms are only getting smarter every day. For example, a new algorithm has reported an accuracy of 84 % for the 50-salads dataset (as compared to the 83%, reported in our original manuscript) [2].  Therefore, it is reasonable not to be worried about extremely poor video segmentation algorithms affecting our framework’s performance.  In case the algorithm has the same bias for all demonstrations, the framework will suffer minimal consequence; since the bias is the same, the feature extractor will identify the most common sub-task related features using all segments from a demonstration pool that has been identified as the same subtask.
>
>     Concern #1b: For example, since the proposed framework uses a maximum entropy model, if most (> 50% or > 70%) of the demonstrations are wrong or imperfect, can the framework still give desired results by finding the most consistent group of sub-tasks?
>
> Clarification: The proposed algorithm analyzes the frequency of spatio-temporal features from correct and incorrect videos in a principled way to infer whether a given video is erroneous or not. If the number of incorrect videos is more than the correct ones, the algorithm will lose this inference power. When learning to infer from labeled data, the number of good quality samples should be more than that of low-quality samples [2]. This is even true for human learning, which is our major inspiration. We would like to present this intuitive example in this regard: imagine you are teaching a 3-year-old how to write an A. You wrote ‘A’ three times and called it an A (equivalent to correct demonstration); You wrote ‘B’ six times and also called it an A (equivalent to incorrect demonstration). The young learner will never know from these 9 demonstrations how to write an A in the correct way. The same is true for a robot that is learning a new task from humans.
>
>     Concern #2: The experimented data are both small in scale as well as very limited in the variations of their scenarios, i.e. 36 videos of human making tea with fixed background and scene. How about demonstration videos in the wild, e.g. making tea in different kitchens by different people?
>
> Clarification: For real-world deployment of LfD-powered robots we need algorithms that can learn from a relatively smaller dataset and do not need a simulator. Our framework is a small step toward this goal. The 50-salad is a benchmark dataset and is being used by the computer vision community for activity recognition and similar research. Our algorithm performs well on this benchmark dataset.
> As for our custom-designed tea-making dataset, three different people were involved in creating the videos in a fixed kitchen background (section 4.2.1). For different kitchen environments, the framework will need a few representative videos from each to learn task-relevant features. Since we are not learning from raw pixels in an end-to-end manner and rely on a feature extraction module, the sample requirement of the proposed framework will never be very high (e.g., in the range of hundreds for each task or environment).

---

> > ### Author Response · Authors · 2022-08-19
> > **Response to reviewer K64Q (Continue)**
> >
> >     Concern #3: Though the overall goal of this paper is to deal with learning from demonstration, the core contribution of the proposed approach seems like an optimization approach for matching or finding consensus that uses max entropy and has no learning components in it. It raises the alert about one of the desk-reject criteria of "No learning" on the CoRL 2022 website: https://corl2022.org/instructions-for-authors/.
> >
> > Clarification: To facilitate visual LfD by robots, the proposed framework takes in training data, learns the Lagrange multipliers from spatio-temporal features extracted from the training data, and identifies the incorrect video demonstrations through analyzing entropy.  Removing incorrect videos from the training set improves the performance of the LfD policy, as shown in Section 4.4. Therefore, the proposed framework itself relies on learning from data (i.e., learning the Lagrange multipliers) and facilitates robot learning from data (the LfD policy). In addition to those, learning from data is happening at different stages of perceptual processing, i.e., video segmentation and spatio-temporal feature extraction.
> >
> >
> > [1] Gammulle, H., Ahmedt-Aristizabal, D., Denman, S., Tychsen-Smith, L., Petersson, L. and Fookes, C., 2022. Continuous Human Action Recognition for Human-Machine Interaction: A Review. arXiv preprint arXiv:2202.13096.
> >
> > [2] Natarajan, N., Dhillon, I.S., Ravikumar, P.K. and Tewari, A., 2013. Learning with noisy labels. Advances in neural information processing systems, 26.

---

### Meta-Review · Area_Chair_TcZ7 · 2022-08-14

**Recommendation:** Accept (Poster)
**Confidence:** 4

**Metareview:**

Paper proposes a method to identify sub-optimal demonstrations. The main idea is following:
- Use an off-the-shelf video segmentation algorithm.
- Identify key spatio-temporal features within each segment.
- From what I understand, next task prediction $p(t | s)$ is used to calculate if the sub-tasks are in the right order, which is a proxy for how good of a demonstration a particular video is. I don't quite understand why this is the right thing to do, any clarification over here would be great. Authors also use terms like "segments" / "tasks" / "subtasks" interchangeably -- which is confusing. Further, $s$ seems to be overloaded to mean the current state and sub-task (Appendix D). I would highly recommend authors to use a single name for each term for clarity.

I agree with the reviewers that this is an important problem, and the results are good on the Yumi Robot video demonstrations. Two reviewers vote to accept the paper, and the other two to reject. The primary concerns of the reviewers are:

- Reviewer key5: Technical correctness, unsubstantiated claims, imprecise language.
- Reviewer K64Q: analysis of how performance changes with segmentation errors, would the method work in complex scenes with realistic background.
- Reviewer av9b: Further discussion of related works

I recommend authors to address all the reviewer concerns (both above and others in the reviews) and improve clarity of their method. It will also be good to see an explicit list of limitations -- where would the ranking method fail?

**Post Rebuttal Update**
Authors provided detailed responses to reviewer questions and consequently reviewer key5 switched from weak reject to accept. In my opinion the authors has addressed the majority of concerns and the paper can be accepted. I encourage the authors to make all the edits promised in the rebuttal and be more explicit about limitations in the camera ready version.

---

> ### Author Response · Authors · 2022-08-19
> **Response to Area Chair TcZ7**
>
>     "From what I understand, next task prediction p(t|s) is used to calculate if the sub-tasks are in the right order, which is a proxy for how good of a demonstration a particular video is. I don't quite understand why this is the right thing to do, any clarification over here would be great."
>
> Regarding this concern:  We are focused on learning multi-step sequential (MSS) tasks – e.g., tea-making, preparing dinner table, etc.  Learning the sequence (of motor actions or presence of features) is one nominal way humans learn sequential tasks [1]. There may be other ways but we adopted this strategy for robot learning of MSS tasks. Therefore, understanding whether the sequence of sub-tasks (i.e. sequential presence of different sets of features) in a given video conforms with those in the training set is important for the proposed model to detect erroneous demonstrations. A definition of the correct demonstration is in footnote 3 on the first page of the original manuscript.
>
>     "Authors also use terms like "segments" / "tasks" / "subtasks" interchangeably -- which is confusing. Further, seems to be overloaded to mean the current state and sub-task (Appendix D). I would highly recommend authors to use a single name for each term for clarity. "
>
> We acknowledge that it could be confusing and will revise the manuscript to be consistent in the use of terminology.
>
>       Reviewer key5: Technical correctness, unsubstantiated claims, imprecise language.
>
> The claim has been answered in the reviewer's comments
>
>      Reviewer K64Q: analysis of how performance changes with segmentation errors, would the method work in complex scenes with realistic background.
>
> The claim has been answered in the reviewer's comments
>
>      Reviewer av9b: Further discussion of related works
>
> The claim has been answered in the reviewer's comments
>
> Finally, the manuscript with all promised revisions will be uploaded for the camera ready version.
>
>  [1] S. P Wise and D T. Willingham, Motor Skill Learning, Encyclopedia of Neuroscience, 2009